# Comparative Chloroplast Genomics, Phylogenomics, and Divergence Times of *Sassafras* (Lauraceae)

**DOI:** 10.3390/ijms26157357

**Published:** 2025-07-30

**Authors:** Zhiyuan Li, Yunyan Zhang, David Y. P. Tng, Qixun Chen, Yahong Wang, Yongjing Tian, Jingbo Zhou, Zhongsheng Wang

**Affiliations:** 1College of Life Sciences, Nanjing University, Nanjing 210023, China; 502023300067@smail.nju.edu.cn (Z.L.); zyynju@nju.edu.cn (Y.Z.); 502024300060@smail.nju.edu.cn (Q.C.); 502024300072@smail.nju.edu.cn (Y.W.); mg1830075@smail.nju.edu.cn (Y.T.); mg1830084@smail.nju.edu.cn (J.Z.); 2Centre for Rainforest Studies, The School for Field Studies, Yungaburra, QLD 4884, Australia; dtng@fieldstudies.org

**Keywords:** *Sassafras*, Lauraceae, comparative chloroplast genomics, divergence times, phylogeny of *Sassafras*

## Abstract

In the traditional classification system of the Lauraceae family based on morphology and anatomy, the phylogenetic position of the genus *Sassafras* has long been controversial. Chloroplast (cp) evolution of *Sassafras* has not yet been illuminated. In this study, we first sequenced and assembled the complete cp genomes of *Sassafras*, and conducted the comparative cp genomics, phylogenomics, and divergence time estimation of this ecological and economic important genus. The whole length of cp genomes of the 10 *Sassafras* ranged from 151,970 bp to 154,011 bp with typical quadripartite structure, conserved gene arrangements and contents. Variations in length of cp were observed in the inverted repeat regions (IRs) and a relatively high usage frequency of codons ending with T/A was detected. Four hypervariable intergenic regions (*ccs*A-*ndh*D, *trn*H-*psb*A, *rps*15-*ycf*1, and *pet*A-*psb*J) and 672 cp microsatellites were identified for *Sassafras*. Phylogenetic analysis based on 106 cp genomes from 30 genera within the Lauraceae family demonstrated that *Sassafras* constituted a monophyletic clade and grouped a sister branch with the *Cinnamomum* sect. *Camphora* within the tribe Cinnamomeae. Divergence time between *S. albidum* and its East Asian siblings was estimated at the Middle Miocene (16.98 Mya), *S. tzumu* diverged from *S. randaiense* at the Pleistocene epoch (3.63 Mya). Combined with fossil evidence, our results further revealed the crucial role of the Bering Land Bridge and glacial refugia in the speciation and differentiation of *Sassafras*. Overall, our study clarified the evolution pattern of *Sassafras* cp genomes and elucidated the phylogenetic position and divergence time framework of *Sassafras*.

## 1. Introduction

The genus *Sassafras* J. Presl belongs to the Lauraceae family and includes three extant deciduous trees; *Sassafras albidum* (Laurales: Lauraceae) (Nutt.) Nees., *Sassafras tzumu* (Laurales: Lauraceae) (Hemsl.) Hemsl., *Sassafras randaiense* (Laurales: Lauraceae) (Hayata.) Rehder. This genus was firstly established by the Czech botanist Jan Svatopluk Presl [1] in 1825 who described *S. albidum*, a species endemic to eastern North America. For over eight decades, *S. albidum* was considered monotypic. This view changed in 1907, when British botanist William Hemsley [2] named *S. tzumu* from mainland China. Later, in 1920, the American botanist Rehder [3] named *S. randaiense* in Taiwan, China. The three *Sassafras* species currently present typical East Asia–North America disjunct distribution with ecological and economic importance [4,5]. Regarding their fossil records, Berry [6] first discovered the extinct species *S. hesperia*, which was excavated from Late Miocene deposits in eastern Washington and northwestern Idaho. Poole et al. [7] found another fossil species, *S. oxylon gottwaldii*, with a potential affinity to *Sassafras* from the wood fossils in the Late Cretaceous sediments in the northern region of the Antarctic Peninsula, indicating that *Sassafras* may have Gondwanan origins.

Due to the disputable morphological characteristics of *Sassafras*, such as the flower sexuality, anther orientation (inward or outward), the inflorescence type, and the presence or absence of involucres, the phylogenetic position of *Sassafras* had been controversial in the traditional classification system of Lauraceae. Specifically, in the classification system of Kostermans [8], *Sassafras* is considered to belong to the raceme-false umbel group, which was closely related to the genera *Lindera* and *Litsea* which have involucres and false umbels. In contrast, Van der Werff and Richter [9] believed that *Sassafras* belonged to the raceme-thyrsoid cyme group, which is more closely related to genera such as *Cinnamomum* and *Ocotea* without involucres and with thyrsoid cymes.

With the wide application of molecular phylogenetics, phylogenetic trees constructed by Rohwer, Chanderbali et al., and Rohwer and Rudolph [10,11,12] using different gene fragments found the terminal branches including the *Persea* group and the Laureae-Cinnamomeae clade. Almost all genera with controversial systematic relationships in the Lauraceae family were clustered in this branch, including *Sassafras*. Therefore, it was named the core Lauraceae group. In further phylogenetic studies on the core Lauraceae group, the research results of Li et al. and Nie et al. [13,14] found stronger support for including *Sassafras* into the Cinnamomeae tribe, which is sister to the Laurus tribe. The Cinnamomeae tribe branch and the Laurus tribe branch formed a sister pair. In the phylogenetic tree constructed by Rohde et al. [15] based on nuclear ITS sequences and chloroplast intergenic regions *psb*A-*trn*H and *trn*G-*trn*S sequences, *Cinnamomum* was found to be paraphyletic group and was divided into three lineages. Based on ITS data, *Sassafras* appeared as the sister group of the *Cinnamomum* sect. *Cinnamomum* group, while in the *psb*A-*trn*H and *trn*G-*trn*S phylogenetic trees, it appeared as the sister group of the *Cinnamomum* sect. *Camphora* group. Since *Cinnamomum* is the core genus of the Cinnamomeae tribe, and *Sassafras* showed a sister relationship with different taxonomic groups of *Cinnamomum* based on different sequences, this provided molecular evidence for classifying *Sassafras* into the Cinnamomeae tribe.

Within the last decade, molecular phylogeneticists have started constructing phylogenetic trees using complete cp genome data, as this is more informative than single-gene or multi-gene fragment data. For instance, Song et al. [16] constructed a phylogenetic tree using the published complete chloroplast genome data of 34 species of Lauraceae, which demonstrated unequivocal support for the separation of the Laurus and the Cinnamomeae tribes, and also the classification of *Sassafras* into the Cinnamomeae tribe. Zhao et al. [17] used two *Endiandra* species as the outgroup and conducted a systematic analysis of the complete chloroplast genome data of 30 species in the Persea-Laurus tribe branch of the Lauraceae family, forming three major branches: the Persea-Machilus branch, the Ocotea-Cinnamomum branch, and the Laurus branch. *Sassafras* was located in the Ocotea-Cinnamomum branch, which is sister to the Laurus tribe branch. Jo et al. [18] conducted a phylogenetic analysis of 49 Lauraceae species using 77 protein-coding sequences and four rRNA gene sequences, and obtained six distinct branches: Cryptocaryeae, Neocinnamomeae, Caryodaphnopsideae, Perseeae, Cinnamomeae, Laurus. Among them, *Sassafras* was classified into the Cinnamomeae tribe. Song et al. and Liu et al. [19,20] both divided the Lauraceae family into 9 branches (Hypodaphnideae, Cryptocaryeae, Caryodaphnopsideae, Neocinnamomeae, Cassytheae, Mezilaurus, Perseeae, Cinnamomeae, Laurus) based on plastid genomes, and *Sassafras* was classified into the Cinnamomeae tribe. Yang et al. [21] newly sequenced the plastid genomes of five species based on phytospecimenomics, and the results of phylogenetic analysis also supported the division of the Lauraceae family into nine branches, with *Sassafras* classified into the Cinnamomeae tribe. Song et al. [22] conducted a phylogenetic analysis of 91 species from 29 genera in the Lauraceae family based on mitochondrial genomes, and the results still supported that *Sassafras* and *Cinnamomum* as sister taxa.

Collectively, the currently constructed phylogenetic trees of the Lauraceae family basically support the establishment of the Cinnamomeae tribe, the classification of *Sassafras* into the Cinnamomeae tribe, and the sister-group relationship between the Cinnamomeae tribe and the Laurus tribe [23]. However, in most previous studies, there were problems such as a small number of *Sassafras* samples selected and the failure to include all three species of *Sassafras*. This also led to fewer studies on the phylogenetic relationships among the three species within *Sassafras*, and there was still some uncertainty as to which group, *C.* sect.* Cinnamomum* or *C.* sect. *Camphora*, *Sassafras* was more closely related to.

In addition, estimation of species divergence times is a research hotspot in phylogenetics and biogeography, which not only help us to estimate important parameters such as species differentiation and evolution rates, but also contribute to exploring the influence of geological history, paleoecology and other factors in the process of species evolution [24]. However, research on the divergence time of *Sassafras* is relatively lacking. So far, merely Nie et al. [14] estimated the divergence time of *Sassafras* using ITS and multiple cpDNA markers (*psb*A-*trn*H, *rpl*16, and *trn*L-F); the results showed that *S. albidum* diverged from its two East Asian siblings at 13.80 ± 2.29–16.69 ± 2.52 Mya (mid-Miocene), and the divergence time between *S. tzumu* in mainland China and *S. randaiense* was approximately 0.61 ± 0.75–2.23 ± 0.76 Mya (Pleistocene).

In this study, we first sequenced and assembled the complete chloroplast genomes of ten individuals of *Sassafras* (five individuals of *S. albidum*, three individuals of *S. tzumu*, and two individuals of *S. randaiense*). We aimed to: (1) investigate the mechanism of chloroplast genome evolution of three *Sassafras* species and develop genetic markers such as microsatellites and DNA barcodes for the genus; (2) explore the phylogenetic position of *Sassafras* in the Lauraceae family, the relationship between *Sassafras* and *Cinnamomum*, and the internal phylogenetic relationships of *Sassafras*; and (3) estimate the divergence times of *Sassafras* and infer their biogeography history combined with fossil evidence.

## 2. Results

### 2.1. Characteristics of Sassafras Chloroplast Genomes

The chloroplast (cp) genomes of ten *Sassafras* individuals all displayed a quadripartite structure, consisting of a pair of inverted repeat regions (IRa and IRb), large and small single-copy regions (LSC and SSC) (Figure 1), and ranging from 151,970 bp (*S. randaiense*: TW10) to 154,011 bp (*S. albidum*: B22) (Table 1). The size of the LSC regions ranged from 92,740 bp (*S. tzumu*: BYS) to 93,634 bp (*S. albidum*: B22), the SSC lengths ranged from 18,756 bp (*S. randaiense*: TW10) to 18,885 bp (*S. albidum*: B18), and the IR regions ranged from 20,054 (*S. albidum*: B1) to 20,809 bp (*S. tzumu*: TW3) (Table 1).

The overall GC content of the ten *Sassafras* cp genomes was 39.2% (Table 1). A total of 128 genes were contained in the ten *Sassafras* cp genomes, including 82 protein-coding genes (CDS), 36 transfer RNA (tRNA) genes, 8 ribosomal RNA (rRNA) genes, and two pseudogenes, respectively (Table 2). Among these genes, 20 genes contained a single intron, while two protein-coding genes possessed two introns (Table 2). The gene *rps*12 was trans-spliced; the exon at the 5′ end located in the LSC region, whereas the 3′ exon and intron located in the IR regions. Moreover, the ^ψ^*ycf*1 and ^ψ^*ycf*2 were identified as pseudogenes because of the partial duplication (Table 2). The cp genomes of ten *Sassafras* individuals were deposited in the GenBank (NCBI). The accession numbers are recorded in Table 1.

The cp genomes comparisons of the ten *Sassafras* in mVISTA revealed a high sequence similarity, and the IR regions and coding regions were more conserved than the LSC region, SSC region, and non-coding regions (Appendix A). In addition, the MAUVE alignment with the algorithm of progressive MAUVE based on the ten *Sassafras* cp genomes showed only one locally collinear block between all analyzed cp genomes and all of the genes exhibited the same and consistent sequence order and no gene re-arrangements or inversion events were detected in these genomes (Figure 2).

The connection sites of various regions of the 10 cp genomes of *Sassafras* were visualized using the IR-Scope online tool (Figure 3). The results showed that there were certain differences in the lengths of the LSC and SSC regions of the chloroplast genomes of 10 individuals from the three *Sassafras* species. The length of the LSC region was 93,534–93,600 bp in *S. albidum*, 92,740–92,753 bp in *S. tzumu*, and 92,772 bp in *S. randaiense*. The length of the SSC region was 18,867–18,885 bp in *S. albidum*, 18,854–18,813 bp in *S. tzumu*, and 18,756 bp in *S. randaiense*. Correspondingly, the length of the IR region was 20,054–20,750 bp in *S. albidum*, 20,096–20,774 bp in *S. tzumu*, and 20,131–20,809 bp in *S. randaiense*.

The boundaries of LSC/IRb (JLB), SSC/IRa (JSA), and IRa/LSC (JLA) were highly conserved, with only minor changes. The boundary of IRb/SSC (JSB) changed significantly. The *ndh*F gene was only 8 bp located in the IRb region in *S. randaiense*, and was only located in the SSC region in *S. albidum* and *S. tzumu*. In addition, the *psb*A and *trn*H genes were located in the LSC region, and the distance from the *trn*H gene to JLA was 1 bp in all cases. The cp genomes of *Sassafras* all had complete *ycf*1 and pseudo *ycf1*^(ψ^*ycf*1) genes. The length of the ^ψ^*ycf*1 gene located in the IRb region was 1382 bp (B12) to 1408 bp (BYS), and only the ^ψ^*ycf*1 in *S. randaiense* was relatively far from the JSB boundary.

The comparison results of codon preference indexed by the value of the relative synonymous codon usage (RSCU) showed that the sequences of *Sassafras* plants had 64 shared codons (Appendix A). Among them, twenty-one codons had RSCU > 1.3 (nine ended with A and twelve ended with T), six codons had RSCU between 1.2 and 1.3 (four ended with T, one ended with A, and one ended with G), and four codons had RSCU between 1.0 and 1.2 (three ended with A and one ended with C). The RSCU values of two codons were equal to 1, which were ATG and TCG. The RSCU values of 31 codons were less than 1, and 28 of them ended with G or C (Appendix A). These results indicated that codons ending with T/A had a relatively high usage frequency in the chloroplast genomes of *Sassafras* plants.

### 2.2. Enrichment of Chloroplast DNA Genetic Resources of Sassafras

A total of 500 dispersed repeat sequences in the cp genomes of *Sassafras* (Appendix A), including 169 forward repeat sequences (f) (33.8%), 108 reverse repeat sequences (r) (21.6%), 21 complementary repeat sequences (c) (4.2%), and 202 palindromic repeat sequences (p) (40.4%). A total of 672 chloroplast SSRs were detected in the chloroplast genomes of *Sassafras* by MISA (Appendix A). The number of SSRs in a single cp genome ranged from 66 (*S. randaiense*: TW3) to 69 (*S. albidum*: B17). Among them, the nucleotides of mononucleotide SSRs (P1, 60.27%) were mainly composed of short A/T repeat sequences, and only a few G (2)/C (4) tandem repeat sequences were present in *S. albidum*. The dinucleotide (P2, 9.52%) types were AT, TA, GA, and TC, and only *S. albidum* had no TC. The trinucleotide (P3, 1.34%) types were TAT, AAT, and AAT only appeared 1 time in B1, and *S. tzumu* had no trinucleotides. The tetranucleotide (P4, 5.65%) types were TTTA, AAAT, AATG, and TTTC, and *S. randaiense* lacked one TTTC. The pentanucleotide (P5) only appeared 1 time in B12, with the type AATAA. No hexanucleotides were found. There were 155 complex nucleotide repeat sequences Pc, accounting for 23.07% of the total number of repeats. The chloroplast SSRs of *Sassafras* were mainly located in the LSC region (79.3%), and the proportions in the SSC (14.7%) and IR (5.95%) regions were relatively small.

Among the 10 individuals of *Sassafras*, the total average Pi value of 106 gene fragments was 0.002763. The average value of the gene-coding region was 0.00159, the average value of the gene intron region was 0.00222, and the average value of the intergenic region was 0.004678. Only four fragments were in the intergenic region, *trn*H-*psb*A and *pet*A-*psb*J were located in the LSC region, and *ccs*A-*ndh*D and *rps*15-*ycf*1 were located in the SSC region. Among them, the *pet*A-*psb*J fragment had the highest Pi value among all fragments (Pi = 0.03081) (Figure 4 and Appendix A).

### 2.3. Phylogenetic Relationships 

*We used 106* cp genomes of 99 species from 30 genera in the Lauraceae family, as well as the cp genomes of *Calycanthus* (*C. floridus* and *C. chinensis*) as the outgroups. Phylogenetic trees were constructed based on two datasets (the whole cp genomes and CDS sequences) using the maximum likelihood method (ML) and the Bayesian method (BI). Results showed that the topological structures of the phylogenetic trees constructed with the complete cp genome sequences (Appendix A) and the CDS datasets (Appendix A) were basically the same. The Lauraceae family can be divided into nine branches: the Laurus-Neolitsea branch, the Cinnamomum-Ocotea branch, the Machilus-Persea branch, the Mezilaurus branch, the Caryodaphnopsis branch, the Neocinnamomum branch, the Cassytha branch, the Beilschmiedi-Cryptocarya branch, and the Hypodaphnideae branch.

The minor differences between the complete cp genome-based tree and the CDS-based tree mainly focused on the topological structure of the Cinnamomeae tribe (including *Sassafras*). Specifically, the Cinnamomeae tribe in our phylogenetic trees consist of the genera *Sassafras*, *Cinnamomum*, *Ocotea*, *Nectandra angustifolia*, and *Licaria capitata*. Among them, *Sassafras* formed a monophyletic group, and the three species within *Sassafras* were also monophyletic taxa. *S. albidum* was at the base of *Sassafras* (BS = 100, PP = 1), *S. tzumu* and *S. randaiense* were sister species (BS = 100, PP = 1, Figure 5). The differences were as follows: in the phylogenetic tree constructed with the complete cp genome sequences (Figure 5a), *Licaria capitata* and *Ocotea bracteosa* formed the base of the Cinnamomeae tribe (BS = 100, PP = 1). The remaining seven *Ocotea* species formed a monophyletic group and were sister species to *Nectandra angustifolia* (BS = 100, PP = 1). *Cinnamomum* was a polyphyletic group and could be divided into three branches: Clade 1, Clade 2, and Clade 3 (Figure 5a). Clade 1 was a sister group to the *Ocotea* and *Nectandra angustifolia* complex (BS = 100, PP = 1). Clade 2 was a sister group to the *Sassafras* branch (BS = 100, PP = 1). Clade 3 was a sister group to the branch formed by Clade2 and *Sassafras* (BS = 100, PP = 1) (Figure 5a).

In contrast, in the cp CDS-based phylogenetic tree (Figure 5b), Ocotea complex (*Ocotea*, *Nectandra*, and *Licaria*) formed the base of the Cinnamomeae tribe and was sister species to the branch formed by *Cinnamomum* and *Sassafras* (BS = 100, PP = 1). *Nectandra* was at the base of the Ocotea complex. *Cinnamomum* was a paraphyletic group and could be divided into two branches: Clade 4 and Clade 5 (Figure 5b). Clade 4 was sister to the *Sassafras* (BS = 100, PP = 1). Clade 5 was a sister group to the branch formed by Clade 4 and *Sassafras* (BS = 100, PP = 1).

### 2.4. Divergence Time of Sassafras

The results of divergence time estimation (Figure 6 and Appendix A) showed that the Cinnamomeae tribe originated at Early Oligocene (30.92 Mya; 95% HPD = 20.85–45.84 Mya) and diverged from the Laurus tribe at Middle Oligocene (28.16 Mya; 95% HPD = 16.19–45.84 Mya). The *C.* sect.* Cinnamomum* and the *C.* sect. *Camphora* + *Sassafras* complex separated at Late Oligocene (24.51 Mya; 95% HPD = 13.13–45.77 Mya). The divergence between *Sassafras* and the *C.* sect. *Camphora* group occurred at Early Miocene (20.74 Mya; 95% HPD = 10.43–41.27 Mya). *S. albidum* diverged from the two East Asian siblings at Middle Miocene (16.98 Mya; 95% HPD = 8.54–41.27 Mya), and the divergence time between *S. tzumu* and *S. randaiense* was at the Pleistocene epoch (3.63 Mya; 95% HPD = 1.37–9.61 Mya).

## 3. Discussion

### 3.1. Chloroplast Evolution and Development of Genetic Resources for Sassafras

The comparative cp genomes of three *Sassafras* siblings revealed that their genome structures, overall gene arrangement, gene and GC content, and codon usage bias were highly conserved. The expansion/contraction of IR region (such as the size variation of the *ycf*2 gene) is the primary cause of cp genome length variation in *Sassafras* (151,970–154,011 bp). This phenomenon is common in angiosperms. For example, Xiao and Ge [25] found that the larger genome of *Cinnamomum chartophyllum* was caused by the expansion of the IR region.

In addition, abundant genetic resources were detected and developed from the ten cp genomes of *Sassafras* (Appendix A; Figure 1, Figure 2, Figure 3 and Figure 4). Specifically, a total of 500 dispersed repeat sequences and 672 cpSSRs were enriched for *Sassafras*. These repeat sequences and cpSSR molecular markers developed in our study will be useful for population genetics and evolutionary studies of the genus *Sassafras* as well as the molecular marker-assisted selection, breeding and conservation of this and related genera in Lauraceae [26]. Chloroplast DNA molecular markers (divergence hotspot regions or DNA barcodes) have been extensively used for research on plant population genetics, phylogeny and phylogeography [26,27,28]. In this study, four hypervariable region fragments, *ccs*A-*ndh*D, *trn*H-*psb*A, *rps*15-*ycf*1, and *pet*A-*psb*J (Pi > 0.01), were found in *Sassafras* as DNA barcodes.

### 3.2. Phylogenetic Insights

Both the complete cp genome-based and the CDS-based phylogenetic trees in our study strongly supported the following topological structures (Figure 5, Appendix A): (1) nine major branches of Lauraceae: the Hypodaphnideae, the Cryptocaryeae, the Caryodaphnopsideae, the Neocinnamomeae, the Cassytheae, the Mezilaurus, the Perseeae, the Cinnamomeae, and the Laureae, which is consistent with the results of Song et al. [19]; (2) the classification of *Sassafras*, *Cinnamomum*, and the *Ocotea–Nectandra–Licaria* complex into the Cinnamomeae tribe, and the sister relationship between the tribes Cinnamomeae and Laureae; (3) *Sassafras* was a monophyletic group, and the three species within *Sassafras* were also monophyletic groups; *S. albidum* was at the base of *Sassafras*, and *S. tzumu* and *S. randaiense* were sister species (BS = 100, PP = 1), which was consistent with the results of Nie et al. [14] who constructed a phylogenetic tree of *Sassafras* using ITS data and multiple markers (*psb*A-*trn*H, *rpl*16, and *trn*L-*trn*F). However, some differences existed in the internal topological structure of the Cinnamomeae tribe between the complete cp genome-based and the CDS-based phylogenetic trees (Figure 5, Appendix A). The contradiction may result from the differences in evolutionary signals between the two types of datasets: the complete cp genome contains hypervariable non-coding regions, and their evolutionary rates are significantly higher than those of the CDS regions.

With respect to the phylogenetic position of *Sassafras*, in the complete cp genome-based phylogenetic tree, *Sassafras* was a sister group to Clade 2, while on the CDS-based tree, *Sassafras* was a sister group to Clade 4. According to Yang’s research [29] on the classification of *Cinnamomum*, the *Cinnamomum* species in Clade 4 and Clade 2 belonged to the *C.* sect.* Camphora* group. Therefore, *Sassafras* and the *C.* sect. *Camphora* were sister groups, forming a monophyletic group (BS = 100, PP = 1). This is consistent with the result of Rohde et al. [15] that *Sassafras* appeared as the sister group of the *C.* sect. *Camphora* group in the phylogenetic tree established based on *psbA*-*trnH* and *trnG*-*trnS* sequences. *Sassafras* shares significant morphological similarities with *C.* sect. *Camphora*, supporting their close affinity, as evidenced by the following shared traits: both have prominent perulate terminal buds with well-developed, tightly wrapped scales, in contrast to the inconspicuous non-perulate buds of *C.* sect. *Cinnamomum*; their leaves are mostly alternate, clustered at branch apices, with pinnate or weakly tri-plinerved venation, differing from the usually opposite/subopposite leaves with typical tri-plinerved venation in *C.* sect. *Cinnamomum*; both are rich in oil cells in wood and leaves, accumulating volatile terpenoids; and their fruits are fleshy drupes borne on shallow cup-shaped receptacles with apically thickened pedicels, showing a higher degree of morphological matching in fruit and pedicel characteristics compared to other groups [29].

However, it is still inconsistent with the sister-group relationship between *Sassafras* and *C.* sect. *Cinnamomum* supported by the phylogenetic tree established based on ITS data. This phenomenon of nuclear-cytoplasmic inconsistency may be caused by ancient hybridization, gene introgression, or incomplete lineage sorting, which are common in plants [25].

### 3.3. Divergence Time and Biogeographic History of Sassafras

The estimation of divergence time based on fossil calibration showed that the divergence between *Sassafras* and the *C.* sect. *Camphora* group occurred at 20.74 Mya (95% HPD = 10.43–41.27 Mya), *S. albidum* diverged from its two East Asian species at 16.98 Mya (95% HPD = 8.54–41.27 Mya), and the divergence time between *S. tzumu* and *S. randaiense* was 3.63 Mya (95% HPD = 1.37–9.61 Mya). This is similar to the conclusion of Nie et al. [14], the divergence time between *S. albidum* and the two East Asian species was approximately 13.80 ± 2.29–16.69 ± 2.52 Mya, and the divergence time between *S. tzumu* and *S. randaiense* was approximately 0.61 ± 0.75–2.23 ± 0.76 Mya.

#### 3.3.1. Divergence Between *S. albidum* and Its East Asian Species

The divergence time between *S. albidum* and the East Asian clade was estimated at 16.98 Mya (Mid-Miocene), a timing that aligns with both the genus’ long geological evolutionary history and critical transitions in the Northern Hemisphere temperate flora. Fossilized leaves of *S. hesperia*, a Miocene survivor from western North America, confirmed the continuous presence of North American populations in humid forest habitats until the Miocene [6]. This fossil record spatially and temporally corroborated the intercontinental divergence time inferred from molecular clock dating (16.98 Mya), indicating that drastic environmental changes during the Miocene were key drivers of the modern disjunct distribution pattern.

This divergence time (Mid-Miocene) shares significant commonalities with typical East Asia–eastern North America disjunct species, such as *Liriodendron* (14.15 Mya) and *Phryma leptostachys* (3.68–5.23 Mya) all concentrated within the Miocene (23–5 Mya) [30,31]. This pattern reflects the synergistic effects of the “periodic Bering Land Bridge openings and Climatic Optimum” during this interval. From the perspective of speciation theory, the divergence of *Sassafras* integrated mechanisms of allopatric speciation and niche differentiation: during the early Miocene (23–16 Mya), the decline in CO_2_ concentration from 1500 to 700 ppm [32] triggered the contraction of pantropical flora, forcing temperate-adapted *Sassafras* to diverge ecologically from tropical relatives (e.g., *C.* sect. *Camphora*) by 20.74 Mya.

#### 3.3.2. Divergence Between *S. tzumu* and *S. randaiense*

According to the theory of allopatric speciation, the divergence between *S. randaiense* and *S. tzumu* at the Pleistocene epoch (3.63 Mya) fundamentally represents a product of geographic isolation triggered by land bridge closure. While the main body of Taiwan Island formed 4–5 Mya, the critical driver of population genetic divergence was the Pliocene-Pleistocene transition (5.3–2.6 Mya), during which the East China Sea shelf emerged as a migration corridor during glacial sea level drops, allowing some *Sassafras* populations to colonize Taiwan. Interglacial sea level rises (e.g., periodic flooding of the land bridge after 3.63 Mya) severed gene flow between mainland and island populations. This isolation was amplified by Quaternary glacial-interglacial cycles (2.58–0.01 Mya), eventually leading to the formation of a Taiwanese endemic species through independent evolution, while mainland populations could not re-colonize Taiwan due to long-term geographic isolation [33,34,35]. Notably, compared to the divergence times of *Picea wilsonii* (mainland spruce) and *P. morrisonicola* (Taiwan spruce, 1–2 Mya), and *Acer oliverianum* (mainland five-lobed maple) vs. *A. oliverianum* subsp. formosanum (Taiwan five-lobed maple, 2.91 Mya) [36,37], the divergence of *S. tzumu* and *S. randaiense* occurred during an earlier period of land bridge instability, implying more complete isolation and lower potential for gene flow recovery.

Taiwan’s unique topographic and climatic gradients (e.g., high-elevation habitats in the Central Mountain Range, tropical-subtropical monsoon climate) also imposed niche filtering. Modern *S. randaiense* is restricted to humid forests at 1000–2500 m elevation in central Taiwan, with morphological adaptations to high humidity and foggy environments [38]. In contrast, mainland *S. tzumu* populations inhabit low mountainous in southeastern China, adapted to relatively dry habitats with distinct seasonal temperature variations [39]. Over time, the two lineages developed non-overlapping niches, eliminating the ecological basis for mainland *S. tzumu* to re-colonize Taiwan.

## 4. Materials and Methods

### 4.1. Plant Sampling and DNA Extraction of Sassafras

To extract DNA, fresh young leaflets from 10 Sassafras individuals (5 of *S. albidum*, 3 of *S. tzumu*, and 2 of *S. randaiense*) were collected and preserved in silica gel for drying. Specific sampling details and voucher specimen numbers are provided in Appendix A. Genomic DNA was isolated from these silica-dried leaf samples using a modified CTAB protocol [40]. The integrity and quality of the extracted DNA were evaluated via agarose gel electrophoresis and further verified using an Agilent 2100 Bioanalyzer (Agilent Technologies, Santa Clara, CA, USA). Meanwhile, DNA concentration was quantified with a NanoDrop LITE spectrophotometer (Thermo Fisher Scientific, Wilmington, DE, USA).

### 4.2. Illumina Sequencing, Chloroplast Genome De Novo Assembly and Annotation of Sassafras

High-quality genomic DNA from each of the 10 *Sassafras* individuals was utilized to construct Illumina paired-end (2 × 150 bp) sequencing libraries. These libraries were sequenced on a HiSeq Xten platform (Illumina, San Diego, CA, USA) in a single lane, with the sequencing work conducted at the Beijing Genomics Institute (BGI, Shenzhen, China). For each individual, approximately 10 Gb of raw sequencing data were generated. Clean data were obtained using the NGS QC Tool Kit v.2.3.3, which involved filtering out adapters and low-quality reads with a Q-value ≤ 20. The complete cp genomes of the 10 individuals were de novo assembled using GetOrganelle v.1.7.6.1 [41] with default parameters. Annotation of these cp genomes was performed via GeSeq v.2.0.0 and CPGAVAS2 v.2.1.0 [42,43]. The annotated cp genomes were then submitted to the NCBI GenBank database, with their respective accession numbers listed in Table 1. Circular physical maps of the *Sassafras* cp genomes were constructed using OrganellarGenomeDRAW v.1.3.1 [44], followed by manual adjustments for accuracy.

### 4.3. Comparative Chloroplast Genome Analysis of Sassafras

To determine the sequence divergence levels among *Sassafras* species, we compared the 10 cp genomes using the mVISTA program (https://genome.lbl.gov/vista/index.shtml, accessed on 1 July 2024) [45]. The analysis was performed under the Shuffle-LAGAN mode, with the annotated cp genome of *S. tzumu* (GenBank ID: NC045268) serving as the reference.

To explore genome-wide evolutionary dynamics and structural variations across the 10 *Sassafras* individuals, MAUVE v.1.1.1 (https://darlinglab.org/mauve/mauve.html, accessed on 3 July 2024) [46] was utilized to detect key evolutionary events in multiple sequence alignments, including gene loss, duplication, rearrangement, and translocation. Additionally, IRscope (https://irscope.shinyapps.io/irapp/, accessed on 3 July 2024) [47] was employed to visualize the overall cp genome structure and track size variations at the boundary regions between inverted repeat (IR), small single copy (SSC), and large single copy (LSC) regions.

Codon usage patterns and relative synonymous codon usage (RSCU) values [48] were calculated for all protein-coding genes in the 10 cp genomes using CodonW v.1.4.2 (http://codonw.sourceforge.net/, accessed on 8 July 2024) [49]. For this analysis, genes shorter than 300 bp and duplicated genes were excluded. Prior to computation, two non-degenerate unique codons (AUG and UGG) and three stop codons (TAA, TAG, TGA) were removed to ensure accuracy.

### 4.4. Identification of cp Microsatellites, Repeats, and DNA Barcodes of Sassafras

To detect chloroplast microsatellite markers (simple sequence repeats, SSRs) in the cp genomes of the 10 *Sassafras* individuals, we employed the MIcroSAtellite (MISA) perl script [50] with specific parameter settings: 10 repeat units for mononucleotide SSRs, 6 for dinucleotide SSRs, and 5 for tri-, tetra-, penta-, and hexa-nucleotide SSRs.

REPuter [51] was utilized to quantify four types of dispersed sequence repeats (complement, forward, reverse, and palindromic repeats) in the 10 cp genomes, with parameters set to a minimum repeat length of 50 bp and a Hamming distance of 8.

For identifying potential DNA barcodes of *Sassafras*, DnaSP v.6.0 [52] was used to calculate nucleotide variability (π) in both coding regions and non-coding regions. Prior to this, sequences were aligned using MAFFT v.7 [53], and only those with an aligned length >200 bp and ≥1 mutation site were included. The resulting π values were visualized using R v.4.0.2.

### 4.5. Phylogenetic Analysis

To clarify the phylogenetic placement of *Sassafras* within the Lauraceae family, we analyzed 10 *Sassafras* individuals alongside 96 published complete chloroplast (cp) genomes from 30 Lauraceae genera (retrieved from NCBI; detailed information in Appendix A). *Calycanthus floridus* and *C. chinensis* were designated as outgroups, and sequence alignment was performed using MAFFT v.7 [54].

Phylogenetic trees were constructed via maximum likelihood (ML) and Bayesian inference (BI) methods, based on two datasets: the complete cp genome and concatenated sequences of 43 common protein-coding genes (CDS). These 43 genes were identified from the 108 cp genomes using Phylosuite [55,56], which was also employed for multiple sequence alignment and sequence concatenation to generate the CDS dataset.

Model selection and parameterization were based on BIC values calculated by jModelTest v.2.1.10 [57,58]. The ML tree was built with IQ-TREE (http://www.iqtree.org/) under the GTR + R3 + F model, with 1000 ultrafast bootstrap replicates [59]. For the BI tree, MrBayes (http://nbisweden.github.io/MrBayes/, accessed on 11 July 2024) was used with the GTR + I + G + F model [60]; Markov Chain Monte Carlo (MCMC) runs lasted 1,000,000 generations (sampled every 1000 generations), with the first 25% of trees discarded as burn-in. The remaining trees were used to construct a 50% consensus tree, with posterior probabilities (PPs) estimated from the top 25% best-scoring trees. Convergence was confirmed when the average standard deviation of split frequencies fell below 0.01. For the ML analysis, two independent searches were conducted to verify topological consistency, with nodal support assessed via 1000 bootstrap (BS) replicates per run. Finally, the phylogenetic tree topology was visualized using the iTOL online tool (https://itol.embl.de/, accessed on 28 July 2024) [61].

### 4.6. Divergence Time Estimation

We used the BEAST v.2.6.3 to estimate the divergence time of *Sassafras* species based on the dataset composed of the concatenated sequences of 43 shared CDS [62]. For calibrating branch divergence times within the Cinnamomeae clade, we adopted the same fossil markers as reported by Xiao and Ge [25]: we used the fossil flower of *Virginianthus calycanthoides* to calibrate the crown age of the Lauraceae family to 107.1 ± 0.5 Mya, the *Neusenia tetrasporangiata* Eklund fossil was used to calibrate the crown node age of the Neocinnamomum-Caryodaphnopsis core Lauraceae branch to 83 ± 1 Mya, and the *Machilus maomingensis* fossil was used to calibrate the stem node age of *Machilus* to 33.7 ± 1 Mya. The dataset file was imported into BEAUti and parameters were set as follows: GTR as the nucleotide substitution model, lognormal relaxed as the molecular clock model, Yule Model as the prior setting. The MCMC method (running 10,000,000 times, sampling every 1000 times) was used to estimate the divergence time, and an xml file was obtained. This file was run in BEAST v.2.6.3, and the resulting log file was checked for convergence using Tracer v.1.7.1 [63]; effective sample size (ESS) ≥200 indicated successful convergence, while ESS <200 required increasing the number of iterations. In Tree Annotator, the first 10% of samples were discarded as burn-in to produce a time-calibrated tree file, which was then visualized using FigTree v.1.4.0.

## 5. Conclusions

Our study first sequenced, assembled and conducted the comparative cp genomics of 10 individuals of three *Sassafras* siblings. We found the whole length of cp genomes of the 10 *Sassafras* had typical quadripartite structure, and conserved gene arrangements and contents. Four hypervariable intergenic regions (*ccs*A-*ndh*D, *trn*H-*psb*A, *rps*15-*ycf*1, and *pet*A-*psb*J) and 672 cpSSRs were identified for the enrichment of genetic resources of *Sassafras*. Phylogenetic analysis based on 106 cp genomes from 30 genera within the Lauraceae family demonstrated that *Sassafras* constituted a monophyletic clade and grouped a sister branch with the *C*. sect. *Camphora* within the tribe Cinnamomeae. Divergence time between *S. albidum* and its East Asian siblings was estimated at the Middle Miocene (16.98 Mya), *S. tzumu* diverged from *S. randaiense* at the Pleistocene epoch (3.63 Mya). Combined with fossil evidence, our results further revealed the crucial role of the Bering Land Bridge and glacial refugia in the speciation and differentiation of *Sassafras*. Overall, our study clarified the evolution pattern of *Sassafras* cp genomes and elucidated the phylogenetic position and divergence time framework of *Sassafras*.

## Figures and Tables

**Figure 1 ijms-26-07357-f001:**
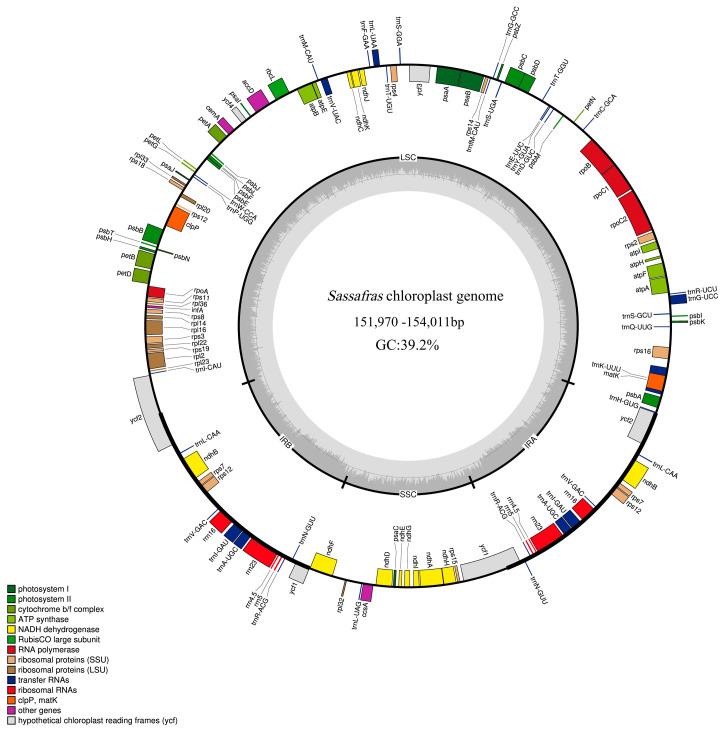
Circular map of chloroplast genomes of *Sassafras* with annotated genes. Genes shown inside and outside of the circle are transcribed in clockwise and counter-clockwise directions respectively. Genes belonging to different functional groups are color-coded. The GC and AT content are denoted by the dark grey and light grey color in the inner circle, respectively. LSC, SSC, and IR are large single-copy region, small single-copy region, and inverted repeat region, respectively.

**Figure 2 ijms-26-07357-f002:**
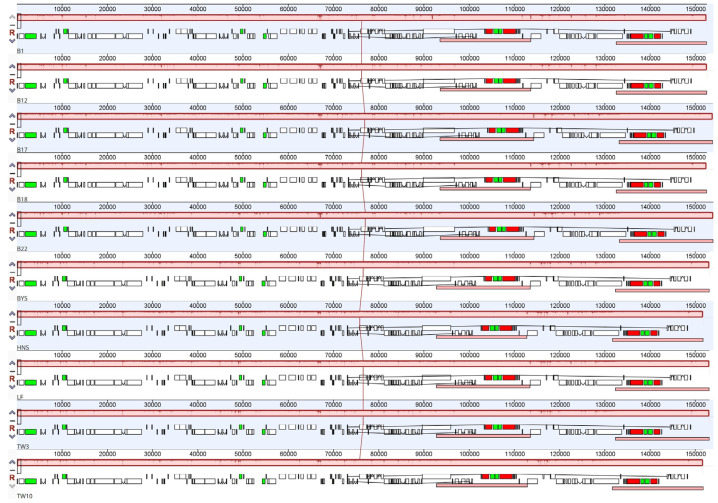
Alignment of ten *Sassafras* chloroplast genomes. Genome of *Sassafras albidum* (B1) is shown at the top as the reference genome. Within each of the alignments, local collinear blocks are represented by blocks of the same color connected by lines.

**Figure 3 ijms-26-07357-f003:**
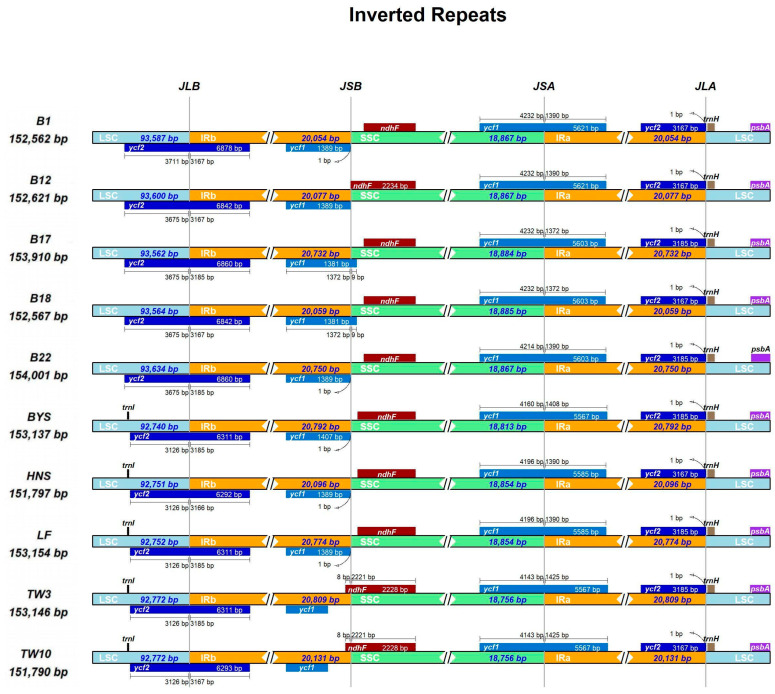
Comparison of the borders of the IR, SSC and LSC regions among ten chloroplast genomes of *Sassafras*. JLB, JSB, JSA, and JLA represent the junctions of LSC/IRb, IRb/SSC, SSC/IRa, and IRa/LSC, respectively.

**Figure 4 ijms-26-07357-f004:**
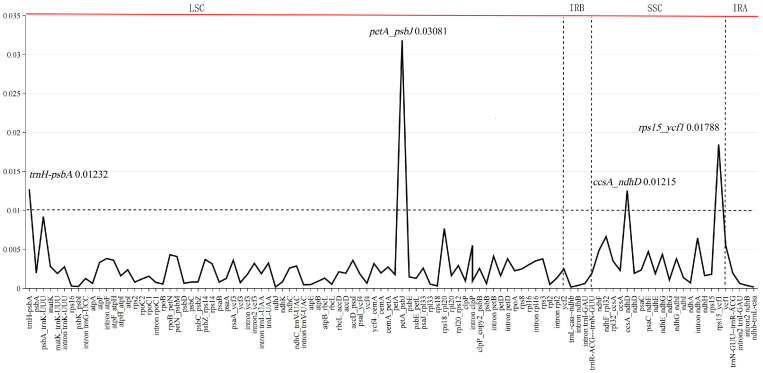
Nucleotide variability (Pi) values of ten individuals of *Sassafras* chloroplast genomes, respectively.

**Figure 5 ijms-26-07357-f005:**
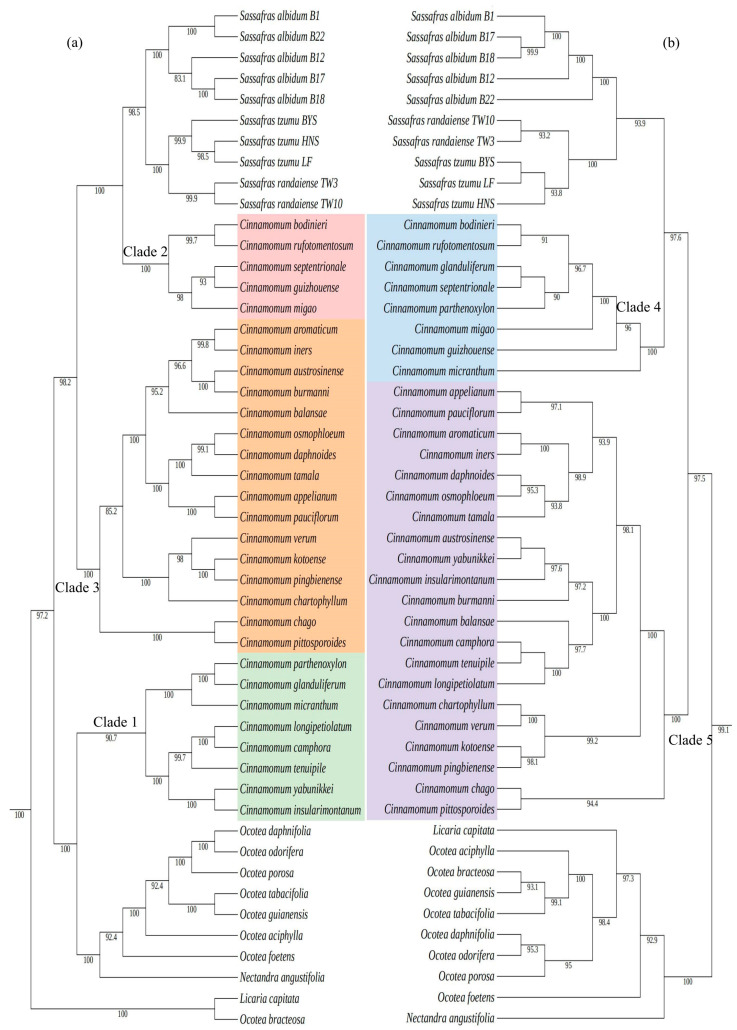
Part of the Cinnamomeae tribe in maximum-likelihood and Bayesian analysis trees constructed based on 106 chloroplast genomes of 99 species from 30 genera in the Lauraceae family. Left (**a**) is the phylogenetic tree constructed with the complete chloroplast genomes, and right (**b**) is the phylogenetic tree constructed with the CDS genes. Numbers above the lines represent ML bootstrap.

**Figure 6 ijms-26-07357-f006:**
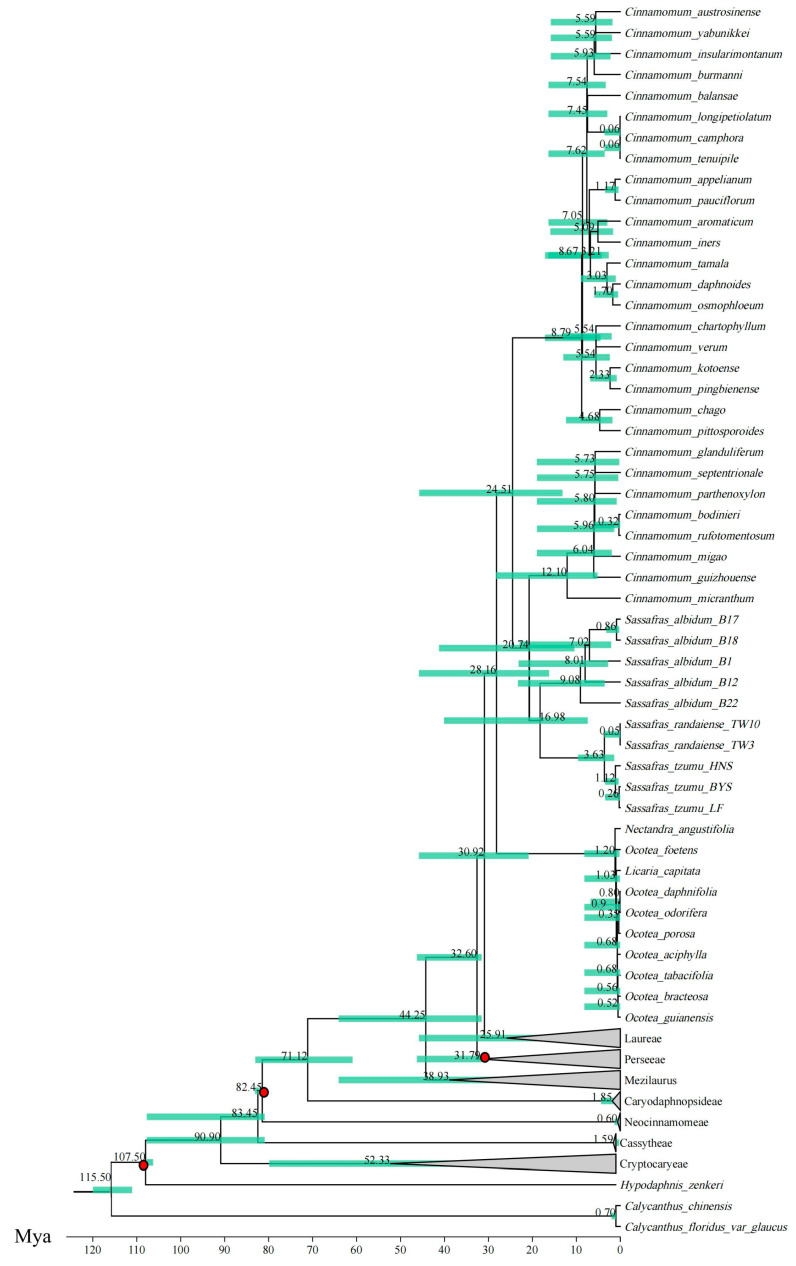
Divergence time estimation using plastid protein-coding genes (PCGs). Green node bars indicate 95% highest posterior distributions; three red circles indicate fossil calibration points.

**Table 1 ijms-26-07357-t001:** Comparison of complete plastid genomes of 10 *Sassafras* individuals.

Species	Collection Number	Genbank ID	Whole Length (bp)	Length of LSC (bp)	Length of IR (bp)	Length of SSC (bp)	Total GC Content (%)	Number of CDS	Number of tRNA	Number of rRNA
*S. albidum*	B1	MW683126	152,562	93,587	20,054	18,867	39.2	82	36	8
*S. albidum*	B12	MW696794	152,621	93,600	20,077	18,867	39.2	82	36	8
*S. albidum*	B17	MW696797	153,910	93,562	20,732	18,884	39.2	82	36	8
*S. albidum*	B18	MW696798	152,567	93,564	20,059	18,885	39.2	82	36	8
*S. albidum*	B22	MW696799	154,001	93,634	20,750	18,867	39.2	82	36	8
*S. tzumu*	BYS	MW696800	153,137	92,740	20,792	18,813	39.2	82	36	8
*S. tzumu*	HNS	MW696801	151,797	92,751	20,096	18,854	39.2	82	36	8
*S. tzumu*	LF	MW696802	153,154	92,752	20,774	18,854	39.2	82	36	8
*S. randaiense*	TW3	MW696808	153,146	92,772	20,809	18,756	39.2	82	36	8
*S. randaiense*	TW10	MW696807	151,790	92,772	20,131	18,756	39.2	82	36	8

Note: LSC, SSC, and IR are large single-copy region, small single-copy region, and inverted repeat region, respectively.

**Table 2 ijms-26-07357-t002:** List of genes in chloroplast genome of *Sassafras*.

Groups of Genes	Names of Genes
Ribosomal RNAs	*rrn*4.5 (×2), *rrn*5 (×2), *rrn*16 (×2), *rrn*23 (×2)
Transfer RNAs	** trn*A*-*UGC (×2), *trn*C*-*GCA, *trn*D*-*GUC, *trn*E*-*UUC, *trn*F*-*GAA, *trn*G*-*GCC, ** trn*G*-*UCC, *trn*H*-*GUG, *trn*I*-*CAU, ** trn*I*-*GAU (×2), ** trn*K*-*UUU, *trn*L*-*CAA (×2), ** trn*L*-*UAA, *trn*L*-*UAG, *trnf*M*-*CAU, *trn*M*-*CAU, *trn*N*-*GUU (×2), *trn*P*-*UGG, *trn*Q*-*UUG, *trn*R*-*ACG (×2), *trn*R*-*UCU, *trn*S*-*GCU, *trn*S-GGA, *trn*S*-*UGA, *trn*T*-*GGU, *trn*T*-*UGU, *trn*V*-*GAC (×2), ** trn*V*-*UAC, *trn*W*-*CCA, *trn*Y*-*GUA
Photosystem I	*psa*A*, psa*B*, psa*C*, psa*I*, psa*J
Photosystem II	*psb*A, *psb*B, *psb*C, *psb*D, *psb*E, *psb*F, *psb*H, *psb*I, *psb*J, *psb*K, *psb*L, *psb*M, *psb*N, *psb*T, *psb*Z
Cytochrome	*pet*A, ** pet*B, ** pet*D, *pet*G, *pet*L, *pet*N
ATP synthase	*atp*A, *atp*B, *atp*E, ** atp*F, *atp*H, *atp*I
Rubisco	*rbc*L
NADH dehydrogenease	** ndh*A, ** ndh*B (×2), *ndh*C, *ndh*D, *ndh*E, *ndh*F, *ndh*G, *ndh*H, *ndh*I, *ndh*J, *ndh*K
ATP-dependent protease subunit P	*** clp*P
Chloroplast envelop membrane protein	*cem*A
Large units	** rpl*2, *rpl*14, ** rpl*16, *rpl*20, *rpl*22, *rpl*23, *rpl*32, *rpl*33, *rpl*36
Small units	*rps*2, *rps*3, *rps*4, *rps*7 (×2), *rps*8, *rps*11, ** rps*12 (×2), *rps*14, *rps*15, ** rps*16, *rps*18, *rps*19
RNA polymerase	*rpo*A, *rpo*B, ** rpo*C1, *rpo*C2
Translational initiation factor	*inf*A
Miscellaneous proteins	*mat*K, *acc*D, *ccs*A
Hypothetical proteins and conserved reading frame	*** ycf*3, *ycf*4, *ycf*1, *ycf*2
Pseudogene	^ψ^ *ycf*1, ^ψ^ *ycf*2

Note: Asterisks (*) before gene names indicate one intron containing genes, and double asterisks (**) indicate two introns in the gene. (×2) indicates genes duplicated in inverted repeat regions (IR). Pseudogene is represented by (^ψ^).

## Data Availability

The data presented in this study are available in the article and Appendix A. The whole chloroplast genome data this study is openly available in GenBank of NCBI at https://www.ncbi.nlm.nih.gov. The accession numbers are recorded in Table 1.

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
