# Peer review of "Comparative Chloroplast Genomics, Phylogenomics, and Divergence Times of *Sassafras* (Lauraceae)"

_ijms, 2025, doi:10.3390/ijms26157357_

Round 1

Reviewer 1 Report

Comments and Suggestions for Authors

The manuscript titled "Comparative chloroplast genomics, phylogenomics, and divergence times of Sassafras (Lauraceae)" addresses the long-debated phylogenetic placement of Sassafras within the Lauraceae family. Through sequencing and analysis of complete chloroplast genomes from 10 Sassafras species, the study reveals conserved genomic structures, key variations in IR regions, and identifies four hypervariable intergenic regions along with 672 microsatellites. Phylogenetic analyses support the monophyly of Sassafras and its close relationship to Cinnamomum sect. Camphora. Divergence time estimates suggest evolutionary events tied to Miocene and Pleistocene periods, likely shaped by the Bering Land Bridge and glacial refugia. The study significantly contributes to understanding Sassafras evolution, phylogeny, and biogeography.

Please revise the following section for clarity and consistency: “Regarding their fossil records, Berry first discovered the extinct species Sassafras hesperia from the Late Miocene excavated in eastern Washington and northwestern Idaho [6]. Poole et al. found another fossil species, Sassafras oxylon gottwaldii, with a potential affinity to Sassafras from the wood fossils in the Late Cretaceous sediments in the northern region of the Antarctic Peninsula, indicating that Sassafras may have Gondwanan origins [7].” Suggested revisions:

  • Add the reference number [7] directly after “Poole et al.”

  • Replace Sassafras oxylon gottwaldii with S. oxylon gottwaldii, and Sassafras hesperia with S. hesperia, as the genus is already established in context.

The results are quite interesting and well presented and illustrated. The discussion is relevant to the problem being addressed, and the materials and methods are described in detail. However, the conclusion should be shortened and more concisely summarized.

Author Response

Thank you very much for taking the time to review this manuscript. Please find the detailed responses below and the corresponding revisions highlighted in the re-submitted files.

Comments1 Please revise the following section for clarity and consistency: “Regarding their fossil records, Berry first discovered the extinct species Sassafras hesperia from the Late Miocene excavated in eastern Washington and northwestern Idaho [6]. Poole et al. found another fossil species, Sassafras oxylon gottwaldii, with a potential affinity to Sassafras from the wood fossils in the Late Cretaceous sediments in the northern region of the Antarctic Peninsula, indicating that Sassafras may have Gondwanan origins [7].” Suggested revisions:Add the reference number [7] directly after “Poole et al.”Replace Sassafras oxylon gottwaldii with S. oxylon gottwaldii, and Sassafras hesperia with S. hesperia, as the genus is already established in context.

Response: Thank you so much for your helpful comment and sorry for our mistake. The changes are as follows:

Regarding their fossil records, Berry [6] first discovered the extinct species S. hesperia, which was excavated from Late Miocene deposits in eastern Washington and northwestern Idaho. Poole et al. [7] found another fossil species, S. oxylon gottwaldii, with a potential affinity to Sassafras from the wood fossils in the Late Cretaceous sediments in the northern region of the Antarctic Peninsula, indicating that Sassafras may have Gondwanan origins.

Comments2 the conclusion should be shortened and more concisely summarized.

Response: Thank you so much for your helpful comment and sorry for our mistake. The changes are as follows:

Our study first sequenced, assembled and conducted the comparative cp genomics of 10 individuals of three Sassafras siblings. We found the whole length of cp genomes of the 10 Sassafras had typical quadripartite structure, conserved gene arrangements and contents. Four hypervariable intergenic regions (ccsA-ndhD, trnH-psbA, rps15-ycf1, and petA-psbJ) and 672 cpSSRs were identified for the enrichment of genetic resources of Sassafras. Phylogenetic analysis based on 106 cp genomes from 30 genera within the Lauraceae family demonstrated that Sassafras constituted a monophyletic clade and grouped a sister branch with the C. sect. Camphora within the tribe Cinnamomeaee. Divergence time between S. albidum and its East Asian siblings was estimated at the Middle Miocene (16.98 Mya), S. tzumu diverged from S. randaiense at the Pleistocene epoch (3.63 Mya). Combined with fossil evidence, our results further revealed the crucial role of the Bering Land Bridge and glacial refugia in the speciation and differentiation of Sassafras. Overall, our study clarified the evolution pattern of Sassafras cp genomes and elucidated the phylogenetic position and divergence time framework of Sassafras.

Reviewer 2 Report

Comments and Suggestions for Authors

The subject of the research presented in the manuscript is well aligned with the scope of this journal. The manuscript requires only minor editorial corrections and clarification of the specific aspects noted.

            Specific comments:

  • Typing errors should be removed i.e. lack of space before brackets with references;
  • Introduction:
  • “This genus was firstly established by the Czech botanist Jan Svatopluk Presl in 1825 by the Czech botanist Jan Svatopluk Presl in 1825 who described..” – repetitions to remove;
  • “……..S. albidum was considered monotypic until 1907, when particular field of research.” – incomprehensible;
  • “References should be numbered in order of appearance and the British botanist William Hemsley named tzumu in mainland China.” – incomprehensible;
  • “Regarding their fossil records, Berry first discovered the extinct species Sassafras hesperia from Late Miocene excavated in eastern Washington and northwestern Idaho[6].….” – it seems that the reference to Berry is missed; further the sentens is unclear and need to be improved;
  • Results; Discussion:
  • The chloroplast……, while…. (table 1).” – why while?
  • Figure 1, 2, 3, 4, 5: better quality should be ensured, as they appear unclear/blurry;
  • Latin names of plant species should be written using Italic font;
  • References should be cited accordingly to the Journal requirements;
  • “…isolation. [33-35]” – to correct;

Author Response

Thank you very much for taking the time to review this manuscript. Please find the detailed responses below and the corresponding revisions highlighted in the re-submitted files.

Comments1 Typing errors should be removed i.e. lack of space before brackets with references;

Response: Thank you so much for your helpful comment and sorry for our mistake.

We have added a space before all the references.

Comments2Introduction:

“This genus was firstly established by the Czech botanist Jan Svatopluk Presl in 1825 by the Czech botanist Jan Svatopluk Presl in 1825 who described.” – repetitions to remove;

“…S. albidum was considered monotypic until 1907, when particular field of research.” – incomprehensible;

“References should be numbered in order of appearance and the British botanist William Hemsley named tzumu in mainland China.” – incomprehensible;

“Regarding their fossil records, Berry first discovered the extinct species Sassafras hesperia from Late Miocene excavated in eastern Washington and northwestern Idaho [6].….” – it seems that the reference to Berry is missed; further the sentens is unclear and need to be improved;

Response: Thank you so much for your helpful comment and sorry for our mistake.

We have changed all the references to specific issues and the result is as follows:

This genus was firstly established by the Czech botanist Jan Svatopluk Presl [1] in 1825 who described S. albidum, a species endemic to eastern North America. For over eight decades, S. albidum was considered monotypic. This view changed in 1907, when British botanist William Hemsley [2] named S. tzumu from mainland China. Later in 1920, the American botanist Rehder [3] named S. randaiense in Taiwan, China. The three Sassafras species currently present typical East Asia-North America disjunct distribution with ecological and economic importance [4,5]. Regarding their fossil records, Berry [6] first discovered the extinct species S. hesperia, which was excavated from Late Miocene deposits in eastern Washington and northwestern Idaho. Poole et al. [7] found another fossil species, S.oxylon gottwaldii, with a potential affinity to Sassafras from the wood fossils in the Late Cretaceous sediments in the northern region of the Antarctic Peninsula, indicating that Sassafras may have Gondwanan origins.

Comments3Results; Discussion:

The chloroplast……, while…. (table 1).” – why while?

Response: Thank you so much for your helpful comment and sorry for our mistake.

Adjust the conjunction according to the relationship between the preceding and following sentences. Here, it means "supplement", so replace it with "and" to ensure logical coherence. The result is as follows:

The chloroplast (cp) genomes of ten Sassafras individuals all displayed a quadripartite structure, consisting of a pair of inverted repeat regions (IR) divided by large and small single-copy regions (LSC and SSC) (Figure 1), and ranging from 151,970 bp (S. randaiense: TW10) to 154,011 bp (S. albidum: B22) (Table 1).

Comments4 Figure 1, 2, 3, 4, 5: better quality should be ensured, as they appear unclear/blurry;

Response: Thank you so much for your helpful comment and sorry for our mistake.

We have reinserted the original high-resolution images of Figures 1 to 5 and corrected the captions.

Comments5 Latin names of plant species should be written using Italic font;

References should be cited accordingly to the Journal requirements;

“…isolation. [33-35]” – to correct;

Response: Thank you so much for your helpful comment and sorry for our mistake.

We have moved the change of reference to before the period and checked the other references.

Reviewer 3 Report

Comments and Suggestions for Authors

The Manuscript ID: ijms-3778598 compares chloroplast genomics, phylogenomics, and divergence times of Sassafras (Lauraceae). The procedures described and materials utilized in their work properly treat the main question addressed by the research on three Sassafras siblings. The related repeat sequences and cpSSR molecular markers developed in their study will be useful for the relevant population genetics and evolutionary studies of the genus Sassafras as well as the molecular marker-assisted selection, breeding and conservation of the studied and related genera in Lauraceae. The study found four hypervariable region fragments, ccsA-ndhD, trnH-psbA, rps15-ycf1, and petA-psbJ (Pi > 0.01) in Sassafras as DNA barcodes. Their hylogenetic analysis based on 106 cp genomes from 30 genera within the Lauraceae family showed that Sassafras constituted a monophyletic clade and grouped a sister branch with the Cinnamomum sect. Camphora within the tribe Cinnamomeaee. Also, the study assessed divergence time between S. albidum and its East Asian siblings at the Middle Miocene (16.98 Mya), S. tzumu diverged from S. randaiense at the Pleistocene epoch (3.63 Mya). Combined with fossil evidence, their data proved the crucial role of the Bering Land Bridge and glacial refugia in the speciation and differentiation of Sassafras. Eventually, this research tried to clarify the evolution pattern of Sassafras cp genomes and to elucidate the phylogenetic position and divergence time framework of Sassafras.

The subject is worth publication and the authors did a good job. Yet, further insights might improve the study and specific improvements should be considered:

  • The authors focused on molecular methods only. They paid little attention to the related differential morphology. When mentioning it, the morphology was stated only as a whole: e.g. “the flower sexuality, anther orientation (inward or outward), the inflorescence type, and the presence or absence of involucres”. They would better combine and consider molecular and morphological methods concurrently for more accurate illumination regarding the phylogeny and systematics of Sassafras in Lauraceae.
  • Each species when reported for the first time in the text should be written in full with Authority and systematics. This is especially important because the phylogenetic position of Sassafras had been controversial in the traditional classification system of Lauraceae, for example, Sassafras albidum (Laurales: Lauraceae) (Nutt.) Nees. And so on.
  • REFs cited (their numbers) should be put down in their proper places in the text, e.g. “the research results of Li et al. and Nie et al. [13,14] found stronger support for including Sassafras into the Cinnamomeae tribe, which is sister to the Laurus tribe.” Instead of “the research results of Li et al. and Nie et al.found stronger support for including Sassafras into the Cinnamomeae tribe, which is sister to the Laurus tribe[13,14].” Likewise in others, e.g. “Song et al. [16] constructed….” Instead of “Song et al. constructed….” And so on.
  • Many typos, misprints, and mistakes were found in the MS and should be corrected, to name but a few:
  1. Abbreviations should be written in full at their first mentioning in the text, then their abbreviated letters may be stated; e.g. inverted repeat regions (IRs), small single copy region (SSC) and large single copy region (LSC)
  2. Such a full mentioning of abbreviations (the above-mentioned indication) should be followed not only in the text but also in the figures and tables as footnotes; that is, each figure and table should be self-explanatory.
  3. In “Figure 1” translating the colors are written in very small fonts, please write larger fonts to be readable.
  4. Berry (empty space?) first discovered the extinct species Sassafras hesperia from Late Miocene excavated in eastern Washington….

Therefore, I would suggest accepting it after major revision.

Author Response

Thank you very much for taking the time to review this manuscript. Please find the detailed responses below and the corresponding revisions highlighted in the re-submitted files.

Comments11 The authors focused on molecular methods only. They paid little attention to the related differential morphology. When mentioning it, the morphology was stated only as a whole: e.g. “the flower sexuality, anther orientation (inward or outward), the inflorescence type, and the presence or absence of involucres”. They would better combine and consider molecular and morphological methods concurrently for more accurate illumination regarding the phylogeny and systematics of Sassafras in Lauraceae.

Response: Thank you so much for your helpful comment and sorry for our mistake. In 3.2, the discussion of the morphological similarity between the genera Sassafras and C. sect. Camphora was added. The changes are as follows:

Sassafras shares significant morphological similarities with C. sect. Camphora, supporting their close affinity, as evidenced by the following shared traits: both have prominent perulate terminal buds with well-developed, tightly wrapped scales, in contrast to the inconspicuous non-perulate buds of C. sect. Cinnamomum; their leaves are mostly alternate, clustered at branch apices, with pinnate or weakly triplinerved venation, differing from the usually opposite/subopposite leaves with typical triplinerved venation in C. sect. Cinnamomum; both are rich in oil cells in wood and leaves, accumulating volatile terpenoids; and their fruits are fleshy drupes borne on shallow cup-shaped receptacles with apically thickened pedicels, showing a higher degree of morphological matching in fruit and pedicel characteristics compared to other groups [29].

Comments2Each species when reported for the first time in the text should be written in full with Authority and systematics. This is especially important because the phylogenetic position of Sassafras had been controversial in the traditional classification system of Lauraceae, for example, Sassafras albidum (Laurales: Lauraceae) (Nutt.) Nees. , Sassafras tzumu (Laurales: Lauraceae) (Hemsl.) Hemsl. and Sassafras randaiense (Laurales: Lauraceae) (Hayata.) Rehder.

Response: Thank you so much for your helpful comment and sorry for our mistake. We added a species note to the first sentence of the first paragraph of the introduction. The changes are as follows:

The genus Sassafras J. Presl belongs to the Lauraceae family and includes three extant deciduous trees; Sassafras albidum (Laurales: Lauraceae) (Nutt.) Nees., Sassafras tzumu (Laurales: Lauraceae) (Hemsl.) Hemsl. and Sassafras randaiense (Laurales: Lauraceae) (Hayata.) Rehder. .

Comments3REFs cited (their numbers) should be put down in their proper places in the text, e.g. “the research results of Li et al. and Nie et al. [13,14] found stronger support for including Sassafras into the Cinnamomeae tribe, which is sister to the Laurus tribe.” Instead of “the research results of Li et al. and Nie et al. found stronger support for including Sassafras into the Cinnamomeae tribe, which is sister to the Laurus tribe[13,14].” Likewise in others, e.g. “Song et al. [16] constructed….” Instead of “Song et al. constructed….” And so on.

Response: Thank you so much for your helpful comment and sorry for our mistake. We have rearranged all the references in the text to ensure that they are placed after the corresponding author's name.

Comments4 Many typos, misprints, and mistakes were found in the MS and should be corrected, to name but a few: Abbreviations should be written in full at their first mentioning in the text, then their abbreviated letters may be stated; e.g. inverted repeat regions (IRs), small single copy region (SSC) and large single copy region (LSC).Such a full mentioning of abbreviations (the above-mentioned indication) should be followed not only in the text but also in the figures and tables as footnotes; that is, each figure and table should be self-explanatory.

Response: Thank you so much for your helpful comment and sorry for our mistake.

We have strictly revised all the formatting errors in the manuscript and are sorry for our own lax formatting. Regarding the specific problem of abbreviation format mentioned, we have added abbreviations notes after the corresponding tables and pictures, for example, table 1.

Comments5In “Figure 1” translating the colors are written in very small fonts, please write larger fonts to be readable.

Response: Thank you so much for your helpful comment and sorry for our mistake.

We have modified Figure 1 by enlarging the font of the color blocks corresponding to the captions to increase the readability of the images.

Comments6Berry (empty space?) first discovered the extinct species Sassafras hesperia from Late Miocene excavated in eastern Washington….

Response: Thank you so much for your helpful comment and sorry for our mistake.

We have removed the spaces and checked the formatting specifications. The sentence has been changed as follows:

Regarding their fossil records, Berry [6] first discovered the extinct species S. hesperia, which was excavated from Late Miocene deposits in eastern Washington and northwestern Idaho.

Reviewer 4 Report

Comments and Suggestions for Authors

In this work, 10 individuals of three genera of Sassafras were sequenced, assembled, and compared. The authors thoroughly characterized the chloroplast genomes and performed a phylogenetic analysis based on 106 cp genomes from 30 genera of the Lauraceae family. This analysis showed that Sassafras is a monophyletic clade and is united in a sister branch with the Cinnamomum section. This work clarified the nature of the evolution of chloroplast genomes of sassafras and revealed the phylogenetic position and time frame of the divergence of Sassafras. The article is written in understandable scientific language and is also thoroughly illustrated with diagrams, phylogenetic diagrams, and tables. Another big plus is that the authors provided a lot of information in the supplementary material and also posted the data in NCBI. I have no comments related to the design of the bioinformatics algorithm, but I have minor comments regarding the layout of the manuscript: the information in the figures needs to be made more readable (especially Figures 2 and 4).

Author Response

Thank you very much for taking the time to review this manuscript. Please find the detailed responses below and the corresponding revisions highlighted in the re-submitted files.

Comments1 I have no comments related to the design of the bioinformatics algorithm, but I have minor comments regarding the layout of the manuscript: the information in the figures needs to be made more readable (especially Figures 2 and 4).

Response: Thank you so much for your helpful comment and sorry for our mistake.

We changed the image generation method for Figure 2 to increase its clarity and readability. For Figure 4, while improving the clarity, we annotated the vertical coordinate, the screening line, and the four selected peak points. 

Round 2

Reviewer 3 Report

Comments and Suggestions for Authors

Accept